# Study of the Effectiveness of Corrosion Resistance Growth by Application of Layered AlN–TiO$_2$ Coatings

**Gulnaz Zh. Moldabayeva [1,*], Artem L. Kozlovskiy [1,2], Erzhan I. Kuldeyev [1], Askar Kh. Syzdykov [1] and Aigul Bakesheva [1]**

[1] Department Petroleum Engineering, Satbayev University, Almaty 050013, Kazakhstan; kozlovskiy.a@inp.kz (A.L.K.); e.kuldeyev@satbayev.university (E.I.K.); a.syzdykov@satbayev.university (A.K.S.); a.bakesheva@satbayeev.university (A.B.)

[2] Laboratory of Solid State Physics, The Institute of Nuclear Physics, Almaty 050032, Kazakhstan

[*] Correspondence: g.moldabayeva@satbayev.university

**Abstract:** The work is devoted to the study of the use of AlN–TiO$_2$ coatings as protective materials against corrosion and natural aging, as well as a rise in wear resistance of the steel surface under long-term mechanical influences. The choice of oxy-nitride coatings obtained by magnetron sputtering by layer-by-layer deposition of layers of aluminum nitride and titanium oxide with layer thicknesses of the order of 50 nm and 100 nm as objects of study is due to their high resistance to external influences, which can have a significant impact on growth in the resistance to degradation processes associated with hydrogenation during the operation of steel structures. During determination of the hydrophobicity/hydrophilicity of AlN–TiO$_2$ coatings, it was found that the applied coatings, regardless of the conditions for their preparation, have hydrophobic properties (the contact angle is ~125–130°), which are preserved both during corrosion tests (except for TiO$_2$ coatings, for which the change in the contact angle after corrosion tests is $\Delta \sim 10°$) and when modeling natural aging processes. During the tribological tests of coating samples, it was found that a growth in the number of spray layers (when alternating them) leads to a rise in wear resistance, both in the case of the initial samples and for samples subjected to corrosion in a model solution of 0.1 M NaCl and when simulating natural aging processes.

**Keywords:** corrosion resistance; hydrophobicity; corrosion protection; coatings; tribological tests

## 1. Introduction

The problem of corrosion of the surface of steel and alloys is one of the most important problems of our time due to the expansion of the number of used pipelines, including those located in places that make them difficult to quickly replace (underground, at the bottom of the seas) [1,2]. Moreover, in this case, the occurrence of emergency situations associated with the failure of certain areas can lead not only to environmental problems (spills, pollution) but also to large economic costs associated with both the need for repair work and downtime [3,4]. The occurrence of build-up and oxides during the operation of steel structures, in particular pipelines, can lead not only to the acceleration of corrosion degradation processes but also to contamination of pumped petroleum products by corrosion by-products. At the same time, the resulting build-up or oxides can lead to an increase in the coefficient of friction, which will lead to an accelerated decrease in wear resistance and an increase in the likelihood of the occurrence of grooves during friction. You should also take into account the fact that corrosion processes can occur not only on the inner surface of pipelines, which leads to a decrease in flow rate due to build-up in the form of oxides or paraffin deposits (when pumping petroleum products), but also from the outside when the surface of pipelines interacts with the atmosphere or sea water (during the operation of pipelines laid on the seabed). In this case, the occurrence of corrosion inclusions in the form of pitting or ulcers can have negative consequences on the stability of pipelines, as well

as their wear resistance [5,6]. It is also important to protect the welding areas of pipelines, which, due to production technology, are most susceptible to degradation due to improper manufacturing or in case of operation in aggressive conditions [7,8].

Today, there are several different technological solutions, the use of which can increase the resistance of materials to external influences as well as protect against negative consequences associated with corrosion and degradation when used in aggressive environments or in sea water. Among the proposed technological solutions, one can highlight the application of protective coatings in the form of thin films obtained using magnetron sputtering [9,10], chemical or electrochemical deposition [11–13], as well as using various technologies for applying polymer coatings. These technological solutions are based on the assumption that the applied coating makes it possible to create a barrier layer that prevents corrosion (oxidation) and also reduces the rate of degradation (since none of the coatings considered today is capable of completely protecting against corrosion processes) [14,15]. Also, the use of such technologies for the application of protective coatings makes it possible to increase wear resistance due to higher hardness indicators, which are determined both by the structural features of the selected compounds for coating and by the methods for their preparation, the variation of which allows one to obtain either nanostructured coatings (increased strength due to size effects) or amorphous films (having high strength indicators) [16–18]. Recently, much attention in the field of research related to the development and use of protective coatings has been paid to technological solutions that make it possible to obtain multilayer or layered coatings, including sequential alternation of layers of various compounds, mainly nitrides or oxides, the use of which can increase wear resistance as well as increasing the resistance of materials to external influences. Interest in such research is aimed at the possibility of creating high-strength wear-resistant coatings, the use of which will significantly increase the service life of materials, especially steel structures and pipelines [19–21].

Based on the above, the aim of the study is to determine the influence of the number of layers of different thicknesses of AlN–$TiO_2$ coatings obtained by magnetron sputtering on the resistance to corrosion and degradation associated with exposure to aggressive environments during long-term tests simulating possible real operating conditions. The choice of these layered AlN–$TiO_2$ coatings is due to the prospects of using these coatings to create hydrophobic surfaces that have not only increased resistance to external influences (friction, wear, and strength) but also the possibility of increasing the resistance of materials to degradation. In turn, the use of layered coating technology (i.e., coatings with alternating layers of aluminum nitride and titanium oxide) to create high-strength protective coatings for steels is due to several works [22–24] which propose the use of multilayer coatings that have high stability to gas swelling during the accumulation of radiation damage, as well as high strength characteristics and wear resistance. The strengthening of such coatings is based on the use of technology for the formation of boundary effects when the presence of alternating layers leads to the creation of additional barriers that prevent the spread of microcracks under external influences, as well as structural degradation associated with swelling during the formation of gas-filled cavities or oxidation. Also, such coatings can play an important role during the operation of steel structures at elevated temperatures (about 400–700 °C), operation at which may be accompanied by accelerated oxidation processes due to thermal diffusion in the material. The choice of coatings based on AlN–$TiO_2$ obtained by magnetron sputtering as objects for research is based on the combination of their properties, which allow their use as one of the heat-resistant protective coatings capable of operating at high temperatures. For example, the combination of strength and thermal insulation properties of AlN coatings, which makes it possible to increase the resistance of materials to corrosion processes, is presented in [25]. In [26], the use of multilayer AlN/Si made it possible to increase the corrosion resistance of 304 steel. The use of nanostructured $TiO_2$ modified in various ways as anti-corrosion coatings is considered in [27,28]. Interest in layered coatings based on various compounds, including oxides and nitrides, in the case of their use as protective materials in recent years has been

quite large. In particular, the use of layered coatings makes it possible to create barrier layers on the surface of steel structures that can reduce the rate of degradation. So, for example, the use of layered coatings based on CrNi, TiAlN/CrNi, CrNi–Al$_2$O$_3$–TiO$_2$, and TiN/TiO$_2$ as anti-corrosion coatings was proposed in [29,30]. Using these coatings, the authors managed to increase the wear resistance and corrosion resistance of materials. Much attention is also paid to the development of compounds such as TiO$_2$/graphene or MXene/EP as protective coatings, which have great prospects when used as anticorrosion or antibacterial coatings [31,32]. In addition to the use of multilayer films or coatings as protective anti-corrosion materials, much attention has been paid recently to the use of these structures as cathode materials for batteries or fuels due to their high resistance to external influences and degradation processes [33,34].

Based on the above, the novelty of the presented research lies in determining the variation in the number of layers of AlN–TiO$_2$ coatings for resistance to corrosion and aging, while maintaining the thickness of the coating. Layered coatings can be used not only to increase corrosion resistance but also to strengthen the surface layers, as well as to increase resistance to external influences.

## 2. Materials and Methods

Coatings based on AlN and TiO$_2$ layers were applied using magnetron sputtering onto the surface of 316 L steel, which is one of the most common grades of steel used in industry. To apply the coatings, the method of high-frequency magnetron sputtering was used, implemented on an Auto 500 installation (Edwards, San Jose, CA, USA) [35]. To deposit aluminum nitride, an aluminum target (K. Lesker, Jefferson Hills, PA, USA) was used; the working gas was a mixture of argon (40%) and nitrogen (60%), and the gas was supplied at a pressure of $5 \times 10^{-3}$ mbar. To deposit titanium oxide layers, a titanium target (K. Lesker, USA) was used; the working gas was a mixture of argon (45%) and oxygen (55%), and the gas was supplied at a pressure of $6 \times 10^{-3}$ mbar. Sputtering was carried out at a discharge power of 250 W. To obtain layered coatings, the cathodes were varied during each subsequent deposition. The thickness was determined using the ellipsometry method. The total thickness of the deposited coatings was about $600 \pm 10$ nm. Figure 1 shows a schematic representation of the studied samples of AlN–TiO$_2$ coatings obtained by magnetron sputtering. In total, 4 types of coating were obtained in the experiment: (1) a single-layer AlN coating with a thickness of about 600 nm; (2) a single-layer TiO$_2$ coating with a thickness of about 600 nm; (3) a coating consisting of 3 layers of AlN and 3 layers of TiO$_2$, each 100 nm thick; (4) a coating consisting of 6 layers of AlN–6 layers of TiO$_2$, each 50 nm thick. Figure 1 shows a schematic representation of the resulting coatings. The choice of 6 and 12 deposition layers is determined by the capabilities of the magnetron installation, as well as the possibility of obtaining layers with a minimum thickness of 50 nm. At the same time, the main goal in selecting spraying conditions was the possibility of obtaining coatings with a thickness of about 0.5–0.6 µm with multiple layers during spraying. The reasons for choosing this method for applying coatings are due to the possibility of scaling this coating method to sufficiently large volumes of sample surfaces. Figure 1e shows an example of a side cleavage of the resulting sprayed coatings, combined with mapping results that reflect the elemental composition of the resulting layers. Moreover, according to the obtained mapping data, the layers are compounds of aluminum with nitrogen and titanium with oxygen, which indicates the purity of the sprayed layers, as well as the absence of mixing effects between the layers.

Determination of the properties of hydrophobicity/hydrophilicity of the surface of coatings depending on the conditions of their application (with variations in the number of layers) was carried out using the contact angle method, the measurement of which was carried out by applying a drop of distilled water to the surface of the sample using the "sessile drop" method, followed by photographic recording of the geometry of the resulting drop on the surface. Using the Image J (v 1.8.0.) code program, the geometry and contact wetting angle were determined.

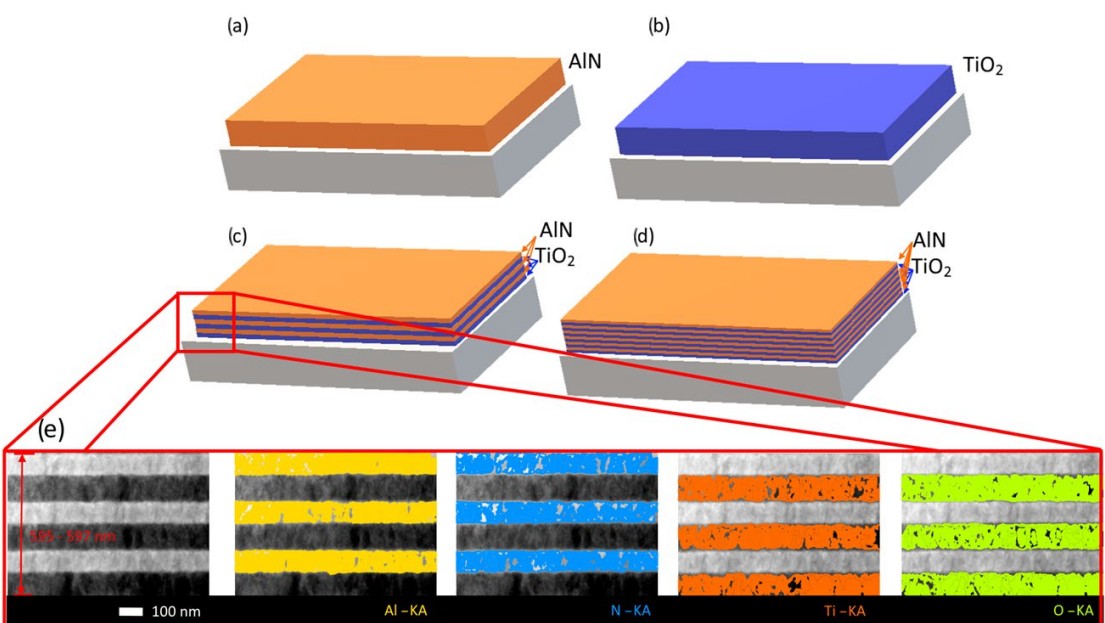

**Figure 1.** Schematic representation of layered AlN–TiO$_2$ coatings: (**a**) 1 layer of AlN; (**b**) 1 layer of TiO$_2$; (**c**) 3 layers of AlN–3 layers of TiO$_2$ with a thickness of 100 nm each; (**d**) 6 layers of AlN–6 layers of TiO$_2$ 50 nm thick each; (**e**) SEM–image of a side chip of 3 layers of AlN–3 layers of TiO$_2$ with the results of elemental analysis in the form of element distribution maps reflecting layer-by-layer deposition.

Determination of the hardness and strengthening effect of the applied coatings depending on the thickness of the applied layers during their alternation was carried out using the indentation method, implemented using a Duroline M1 microhardness tester (Metkon, Bursa, Turkey). A Vickers diamond pyramid was used as an indenter; indentation was carried out under a load of 10 N; the choice of indentation conditions was determined by the possibility of obtaining measurements of coating hardness without affecting the steel surface. The hardening effect was determined by comparing the hardness values of the coatings with the results for single-layer coatings of similar thickness of about 600 nm. Hardness measurements were carried out on the basis of a priori experimental data related to determining the geometry of the indenter prints and their depth during indentation with different loads. The choice of load on the indenter was selected in such a way that the measurement was carried out only at a depth corresponding to the thickness of the coating, without pushing through the substrate.

The adhesive strength of the applied coatings, depending on the number of applied layers, was assessed using the scratching method, implemented on a Unitest framework SKU UT-750 installation (Unitest, New York, NY, USA); an indenter with a Rockwell-type diamond tip (curvature radius of about 200 μm) was used as an indenter. The tests were carried out by moving the indenter along the surface of the coatings at a constant speed of 10 mm/min with a change in load from 0 to 100 N, which in turn made it possible to determine the critical load value at which complete peeling of the coating from the steel surface occurs. Tests to determine changes in adhesive strength as a result of corrosion tests and associated degradation processes were carried out using a similar method; resistance to degradation was assessed as a change in the critical load value in comparison with the initial value before corrosion tests.

The friction coefficient was determined using the standard "ball on disk" scheme with a load on the ball of 50 N at a sliding speed of 0.2 m/s, the number of cycles was about 20,000. An Al$_2$O$_3$ ball with a diameter of 5 mm was used as a counterbody (ball). Determination of the kinetics of degradation (wear resistance) and increase in steel resistance to corrosion due to applied coatings was carried out through a comparative analysis of changes in the

coefficient of friction of the samples in the initial state and after corrosion tests (while in the environment).

Testing of coating samples for corrosion resistance was carried out by placing them in a model solution of 0.1 M NaCl, which was used as a model solution for conducting experiments to determine the resistance of materials to degradation. The main experiments were carried out by placing samples of 316 L steel with layered $AlN–TiO_2$ coatings applied to its surface in a model solution for 500 h (about 21 days), after which the values of the contact angle (determination of surface hydrophobicity/hydrophilicity) and friction coefficient (determining wear resistance) were measured. To determine the influence of the environment on the strength parameters, after every 100 h, measurements of surface hardness and adhesive strength were carried out in order to establish the kinetics of changes in strength characteristics and softening mechanisms of coatings depending on the conditions in the environment.

Also, to determine the possibility of using these coatings not only as protective coatings when used in aggressive environments but also for possible long-term storage of steel materials in air (during operation in open conditions), experiments were conducted to determine the resistance of these coatings to atmospheric corrosion (long-term aging in air). To do this, experiments were carried out to simulate external atmospheric influences by exposure to water vapor at a temperature of 150 °C and a pressure of 2.2–2.3 atm. for 20 h, which in total made it possible to simulate natural aging processes equal to a time period of about 5–7 years. These conditions for conducting aging experiments were chosen in accordance with previously conducted similar experiments [35] related to the demonstration of artificial aging processes, which made it possible to compare the results obtained with the effects of atmospheric conditions on samples during their operation.

## 3. Results and Discussion

Figure 2 shows the results of microphotographs of the contact wetting angle of the studied samples (steel and coatings) in the initial state, after 500 h in a model solution of 0.1 M NaCl, and after modeling natural aging processes (atmospheric corrosion). The presented micrographs characterize changes in the hydrophobic/hydrophilic properties of coatings depending on external influence conditions, which, in comparison with the change data for 316 L steel, characterize the resistance of coatings to degradation under external influences, including corrosion and oxidation. Figure 3 shows SEM images of the surface of coating and steel samples before and after corrosion tests. As can be seen from the presented data, the presence of 316 L steel in model solutions leads to the formation of growths on its surface in the form of salt deposits, as well as oxides. At the same time, for single-layer AlN and $TiO_2$ coatings after corrosion tests, the formation of cavities is observed, indicating the processes of pitting or pitting corrosion, leading to partial destruction of the coating. In the case of layered coatings, such phenomena are not observed, and the main changes are associated only with the formation of small salt deposits on the surface. An analysis of changes in the contact angle is presented in Figure 4 in the form of the results of comparing values before and after various tests (when in an aggressive environment, as well as after modeling natural aging processes).

According to the presented data on the morphological features of the coatings under study in Figure 3, in the case of single-layer AlN coatings, the roughness value before testing is about 30–40 nm. After testing, this value increases to 70–110 nm, and crater-like cavities are observed on the surface, with a depth of about 300–400 nm and a diameter of about 2–3 μm, which indicates ulcerative destruction of the coating. In the case of a $TiO_2$ coating in the initial state, the roughness is about 20–30 nm, and after testing the roughness value is more than 70 nm, while on the surface, as in the case of AlN coatings, the formation of craters and build-ups on the surface of the coating is observed. For $3AlN–3TiO_2$ coatings in the initial state, the roughness is about 40–50 nm, which is comparable to the observed values for AlN coatings. At the same time, for $3AlN–3TiO_2$ coatings after corrosion tests, the roughness practically did not change and was about 45–55 nm. In the case of $6AlN–6TiO_2$

coatings, a decrease in the thickness of the layers leads to the formation of smaller grains, and the roughness value is about 20–30 nm, while this value does not undergo significant changes after corrosion tests.

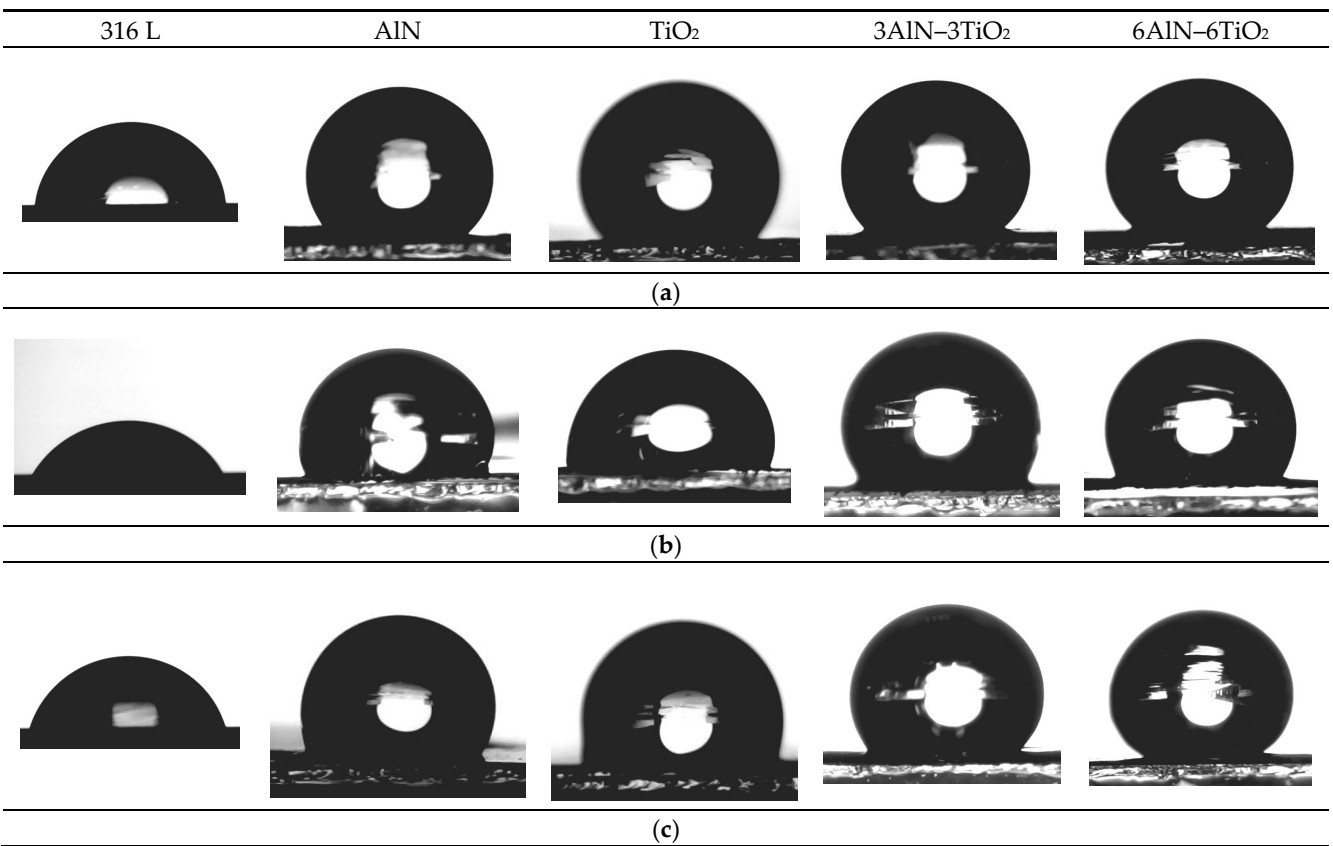

**Figure 2.** Microphotographs of droplets on the surface of the studied steel and coating samples: (**a**) in the initial state; (**b**) after corrosion tests; (**c**) after simulation of natural aging processes.

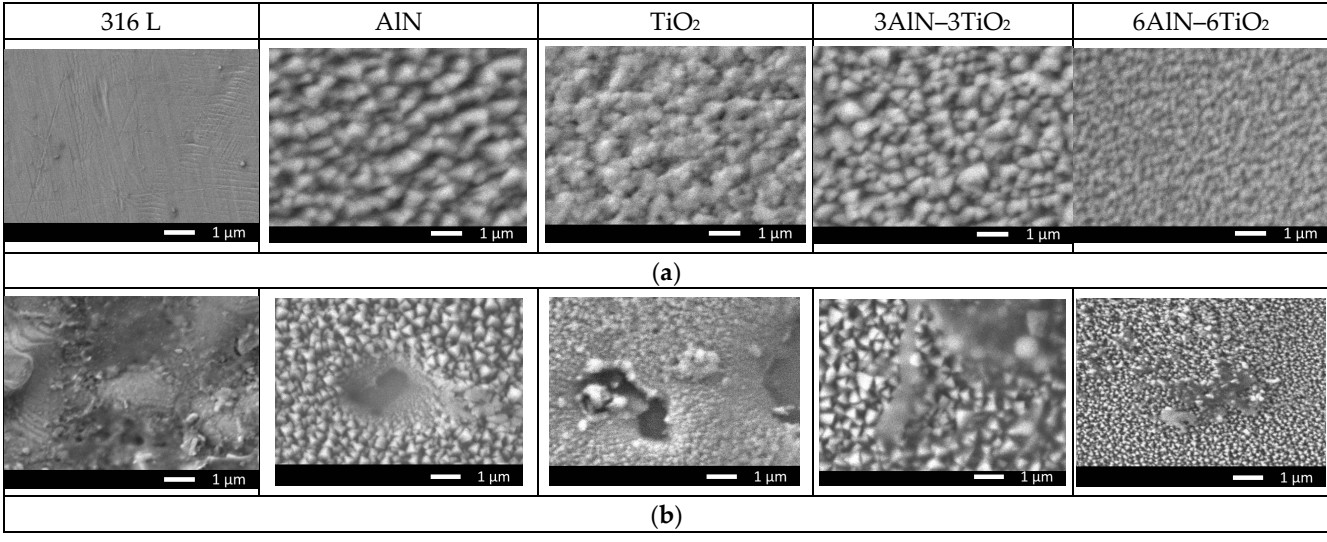

**Figure 3.** SEM images of the studied samples before (**a**) and after (**b**) corrosion tests.

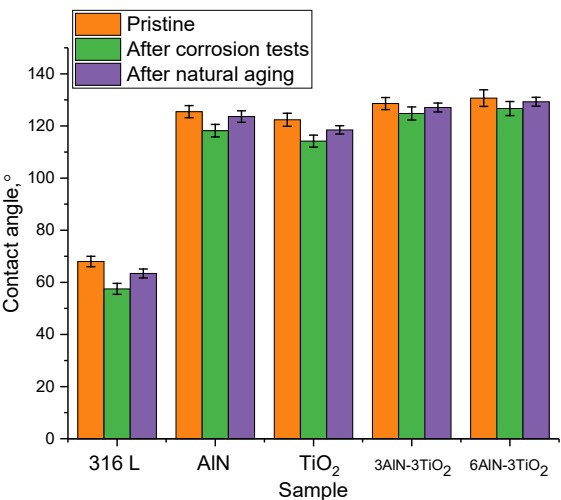

**Figure 4.** Results of changes in the contact angle.

The general appearance of the presented coatings indicates that the deposited AlN–TiO$_2$ coatings in the initial state, regardless of the application conditions (in the case of single or alternating layers), have hydrophobic properties (the contact angle is about 125–130°), while the surface of 316 L steel has all the signs of a hydrophilic surface (the contact angle is about 68–70°). In turn, the presence of hydrophobic properties for AlN–TiO$_2$ coatings can help increase resistance to degradation during corrosion tests, which is clearly visible from the data presented both in micrographs of drops on the surface of the samples (see data in Figure 2), and from the data on changes in the value of the contact angle (see data in Figure 3). In this case, the increase in corrosion resistance is associated with an increase in surface non-wetting, which, with prolonged contact with aggressive media, leads to a decrease in the rate of penetration of Cl$^-$ or OH$^-$ ions into the surface layer and thereby reduces the rate of corrosion, as well as a decrease in strength characteristics. According to the data presented in Figure 3, if 316 L steel is in the aggressive environment of the model solution, the decrease in the contact angle is about 12–14°, which indicates a deterioration in the hydrophobicity of the surface, and as a consequence, an acceleration of degradation processes, which is expressed in a decrease in wear resistance, as well as a decrease in strength characteristics. Moreover, unlike TiO$_2$, for which the decrease in the contact angle after corrosion tests is about 10°, AlN coatings and layered coatings have higher resistance to a decrease in the hydrophobicity of the coatings. It should also be noted that the transition from 6 layers of sprayed coatings to 12 leads to a decrease in the difference in contact angle, which indicates an increase in the resistance of the coating surface to degradation due to an increase in the number of layers, and as a consequence, the number of inter-boundary effects associated with alternating oxide and nitride layers of sputtering. The results obtained are in good agreement with the stability data of multilayer coatings based on oxides and nitrides presented in [36–38], according to which the presence of interlayer boundaries when alternating layers leads to an increase in resistance to degradation under external influences, including gas swelling and embrittlement [36–38].

In the case of samples tested for natural aging (simulation of atmospheric corrosion processes), it was found that there was practically no change in the hydrophobic properties of the coatings (Δ ~ 1–2°), which indicates the high resistance of the coatings to natural aging processes, while for steel 316 L, a decrease in the contact angle was observed, a change in which indicates surface degradation caused by possible oxidation processes during the interaction of water vapor with the surface. For coatings, a small change in the hydrophobicity degree after testing is due to the fact that high values of the contact wetting angle do not allow water vapor to accumulate on the surface of the coating, thereby reducing the likelihood of moisture penetration into the surface layer. A decrease in the

contact angle by 1–2° (which is within the permissible measurement error) is due to the natural degradation of the coating surface during prolonged interaction with water vapor.

Figure 5 reveals the results of assessment of the change in hardness and adhesive strength of the coatings under study before and after corrosion resistance tests (while they were in a model solution).

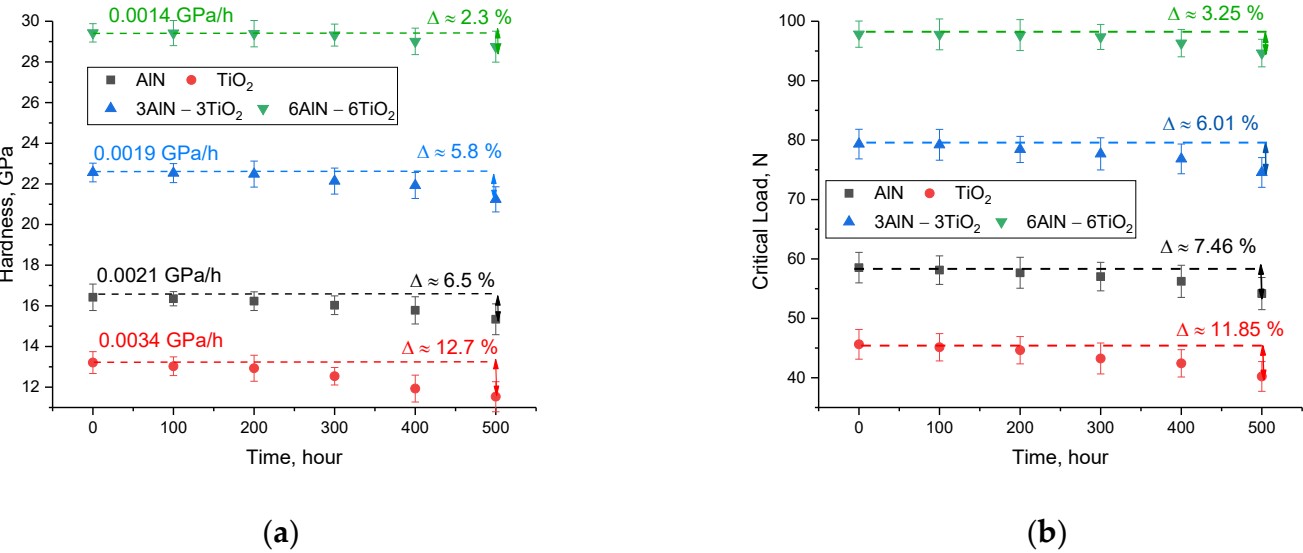

**Figure 5.** Results of changes in the strength characteristics of the coatings under study during simulation of corrosion degradation processes: (**a**) change in hardness; (**b**) change in adhesive strength.

Analyzing these changes in the hardness and adhesive strength of the coatings under study depending on the conditions of their production (i.e., the number of layers), it was found that single-layer $TiO_2$ coatings have the lowest strength characteristics, for which the hardness value is about 13.2 GPa and the adhesive strength is more than 45.6 N. Moreover, in the case of increasing the number of layers for AlN–$TiO_2$ coatings, increasing the number of layers from 6 to 12 leads to a more than 2-fold increase in strength characteristics in comparison with $TiO_2$ coatings. Thus, we can conclude that the use of layer-by-layer spraying of aluminum nitride and titanium oxide leads to the formation of more durable coatings, and an increase in the number of layers increases strength and adhesive strength, which leads to better adhesion of coatings to the surface, as well as increased resistance to mechanical stress.

According to the presented data on changes in strength parameters (hardness and adhesive strength), coatings obtained by layer-by-layer deposition of AlN and $TiO_2$ (in comparison with single-layer coatings of similar thickness) are the most resistant to degradation. Moreover, increasing the number of layers from 6 to 12 leads to a more than twofold increase in resistance to hardness reduction after 500 h of successive testing (the change in hardness is 5.8% and 2.3% for the 3AlN–3 $TiO_2$ and 6AlN–6$TiO_2$ samples, respectively). Moreover, in comparison with changes in the hardness of $TiO_2$ coatings (which showed a maximum decrease of more than 12.5%), the stability for 6AlN–6$TiO_2$ coatings is more than a 5-fold increase in the stability of strength characteristics as a result of degradation. It is also worth noting that the most pronounced decrease in strength characteristics is observed after 300 h of sequential contact with the model solution, which leads to the occurrence of structural distortions, leading to a decrease in hardness. At the same time, the calculated values of the rate of hardness reduction indicate that the formation of layered coatings leads to a more than 1.5–2.0 fold reduction in the rate of degradation of strength parameters.

The assessment results of changes in adhesive strength value (calculation was carried out based on the values of the maximum applied pressure on the indenter before fixing the moment of coating separation) have similar trends of changes as the results of hardness (depending on the time spent in the model solution). According to the data obtained, it

was established that the change in strength parameters has a clear dependence on the time spent in the model solution. At the same time, an increase in the experiment time leads to destabilization of the near-surface layer due to migration processes of the introduction of $Cl^-$ ions into the coatings with a subsequent increase in their concentration, leading to a deterioration in strength properties.

At the same time, the least stable among the coatings in this case are coatings based on $TiO_2$, the analysis of the strength parameters of which showed not only the most pronounced trends in comparison with other types of coatings studied but also lower values of the initial values of the strength parameters (hardness and adhesive strength).

Analysis of the results of changes in the strength parameters of the studied $AlN–TiO_2$ coatings after modeling the processes of natural aging (atmospheric corrosion) showed high resistance of the coatings to the effects of water vapor, and the magnitude of the change in hardness was no more than 0.2%–2% after 20 h of simulation (exposure to water vapor under pressure).

Figure 6 shows the results of tribological tests (changes in the dry friction coefficient) of $AlN–TiO_2$ coatings in the initial state, as well as samples after being in a model solution for 500 h and after tests simulating natural aging processes. The results of tribological tests are presented in the form of changes in the coefficient of dry friction of the surface, determined using the "ball on disk" method. Also, for comparison, to determine the effect of coating on increasing wear resistance (especially during corrosion tests), tribological tests were carried out on 316 L steel, the results of which are also presented in the graphs.

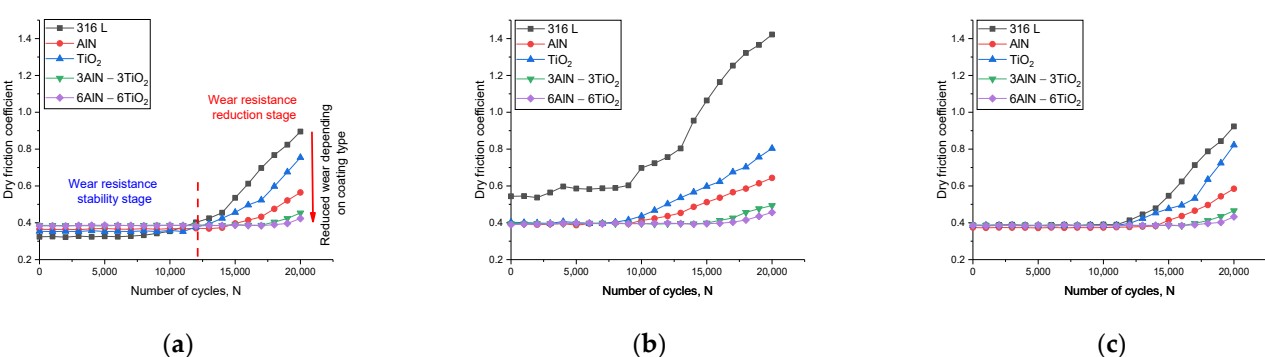

**Figure 6.** Results of tribological tests of the studied samples of coatings and steel: (**a**) in the initial state; (**b**) after corrosion tests; (**c**) after simulation of natural aging processes.

In the case of the original samples (not subjected to testing), it was found that the main changes in the dry friction coefficient are observed after 12,000–13,000 test cycles for single-component coatings, which has good agreement with the data on changes in the dry friction coefficient for steel 316 L. However, despite the fact that wear processes for steel and single-component coatings are observed after the same number of exposure cycles, the value of the specific volumetric wear of the steel surface is significantly higher than for single-component coatings (see data in Figure 7). In the case of two-component $AlN–TiO_2$ coatings, a loss of wear resistance (an increase in the dry friction coefficient) is observed after 16,000–17,000 cycles, while the specific volumetric wear in this case is of the order of $0.2–0.3 \times 10^{-5}$ $mm^3/(N \times m)$, which is an order of magnitude lower than the same value for steel 316 L.

For samples subjected to corrosion tests in a model solution of 0.1 M NaCl, the measured values of the dry friction coefficient after 500 h in the model solution indicate surface degradation, which consists of both an increase in the coefficient of dry friction (and, as a consequence, the value of specific volumetric wear) and a shift in the stage (number of cycles) at which the increase in the coefficient of dry friction begins, indicating the beginning of surface degradation and a decrease in its wear resistance. Moreover, in comparison with the wear resistance data of samples of steel 316 L and $AlN–TiO_2$ coatings, the efficiency of increasing wear resistance is about 1.0–15 times the reduction in the value

of specific volumetric wear. For single-component coatings, the increase in wear resistance of coatings in the case of corrosion tests is about 50%–70% compared to the wear value for steel 316 L.

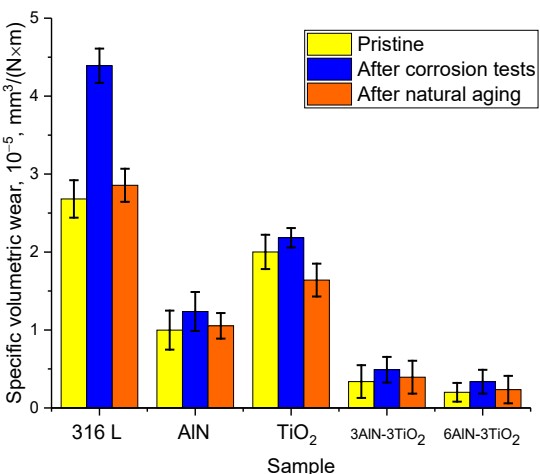

**Figure 7.** Results of a comparative analysis of the specific volumetric wear of the surface of the studied samples depending on the type of external influences.

The results of testing samples when simulating natural aging processes showed a slight increase in wear after testing in comparison with the results for the original samples (not subjected to external influences), which indicates a fairly high resistance of coatings to processes associated with atmospheric corrosion.

In the case of steel structures, the main degradation processes are associated with chemical corrosion processes, which consist of the interaction of the steel surface with the external environment, which is accompanied by a change in the chemical structure, as well as oxidation (formation of oxide inclusions in pores and microcracks). At the same time, the processes of chemical corrosion are accompanied by physical destruction of the surface layer due to the accumulation of oxides, as well as chemical changes in the composition of the surface layer. The mechanisms of surface degradation can be explained by the migration of $Cl^-$ ions deep into the material, due to their good mobility and ability to penetrate microcracks and capillaries (pores) formed in the surface layer of steel during its operation. In this case, the deterioration of strength properties (decrease in hardness, wear resistance and adhesive strength) is a direct consequence of the influence of corrosion processes on changes in the properties of materials, and by assessing the parameters of changes in strength properties, an indirect conclusion can be drawn about the speed of these processes. At the same time, the hydrophobic properties of coatings make it possible to increase resistance to degradation by slowing down corrosion processes, which is most evident for layered coatings, in which an increase in the number of layers leads to an increase in resistance not only to corrosion and wear but also to a decrease in the degradation of strength and adhesive characteristics. The use of AlN–$TiO_2$ coatings obtained by layer-by-layer spraying of 6 or 12 layers leads to an increase in the wear resistance of the surface of 316 L steel, as well as resistance to corrosion and degradation in aggressive environments (model solution 0.1 M NaCl), and in the case of modeling natural aging processes, the use of these coatings makes it possible to maintain wear resistance and strength parameters at a level acceptable by measurement error (the reduction in indicators is no more than 0.5%–2% compared to the original data), which indicates the high resistance of these coatings to processes characteristic of atmospheric corrosion (during natural aging).

Figure 8 shows a schematic representation of the corrosion processes of coating samples when interacting with an aggressive environment. In the case of single-layer coatings, for which the formation of cavities characteristic of ulcer formations associated with interaction with $Cl^-$ ions was observed, degradation is accompanied by both the formation of

growths and partial destruction of the surface. Moreover, for steel 316 L, these processes proceed much faster due to the formation of oxides in the form of iron oxide, which leads not only to the formation of build-up but also to surface degradation in the form of microchips and cracks. In the case of layered coatings, the presence of interlayer boundaries leads to inhibition of the processes of propagation of ulcer formations in depth, which in turn leads to the fact that the cavities formed on the surface of layered coatings have very small sizes.

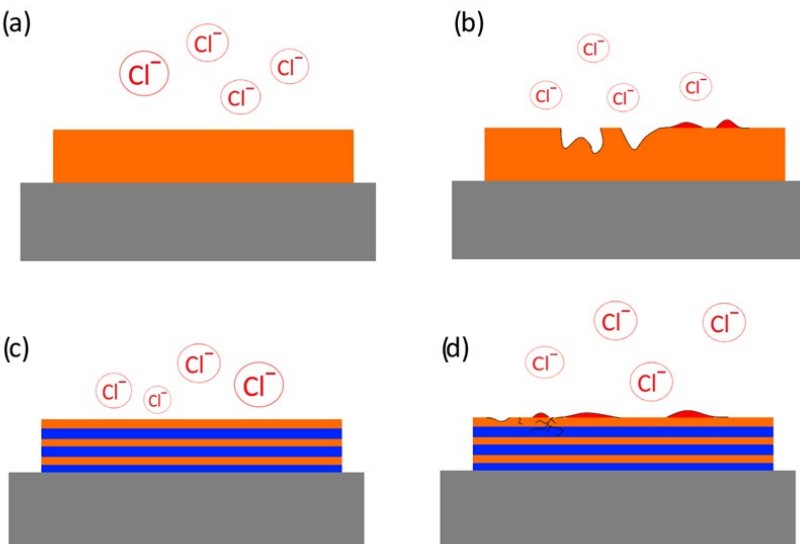

**Figure 8.** Schematic representation of the corrosion processes of coating surfaces when interacting with an aggressive environment: (**a**) single-layer coatings at the beginning of testing; (**b**) single-layer coatings after testing; (**c**) multi-layer coatings at the beginning of testing; (**d**) multi-layer coatings after testing.

## 4. Conclusions

Experiments to determine the effectiveness of using AlN–TiO$_2$ coatings as protective materials to increase the corrosion resistance of steel structures, as well as increase wear resistance under long-term mechanical influences (friction and aging), were conducted. It has been established that alternating layers of AlN–TiO$_2$ coatings leads to an increase in strength parameters of more than 30% (comparing coatings of 6 layers and 12 layers), as well as an increase in adhesive strength of more than 20%. At the same time, the hardening effect is due to the presence of interlayer boundaries (when alternating layers), which not only increase hardness by increasing crack resistance (barrier effect) but also increase adhesion to the steel surface (increasing the maximum load required to tear the coating off the surface). It has been determined that the increase in resistance to wear under mechanical influence in the case of corrosion tests is due to the effects of alternating layers of AlN–TiO$_2$ coatings, which results in the creation of additional barriers to the formation of corrosion ulcer inclusions in the near-surface layer associated with the processes of interaction of the environment with the material. It has been determined that a growth in the alternation of layers from 6 to 12 in AlN–TiO$_2$ coatings leads to an elevation in wear resistance due to the preservation of hydrophobic properties that prevent the acceleration of chemical corrosion processes that occurs during prolonged exposure to an aggressive environment (model solution of 0.1 M NaCl). As a result of modeling the aging processes, it was found that the use of layered coatings can significantly reduce the rate of natural aging of the surface, as well as maintain strength parameters practically unchanged.

The obtained experimental results will be used in the future in the development of technological solutions related to increasing the service life of steel structures operating in difficult conditions, including long-term interaction with aggressive environments.

**Author Contributions:** Conceptualization, G.Z.M., A.L.K., E.I.K., A.K.S. and A.B.; methodology, G.Z.M., A.L.K., E.I.K., A.K.S. and A.B.; formal analysis, G.Z.M., A.L.K., E.I.K., A.K.S. and A.B.; investigation, G.Z.M., A.L.K., E.I.K., A.K.S. and A.B.; resources, G.Z.M., A.L.K., E.I.K., A.K.S. and A.B.; writing—original draft preparation, review, and editing, G.Z.M., A.L.K., E.I.K., A.K.S. and A.B.; visualization, G.Z.M., A.L.K., E.I.K., A.K.S. and A.B.; supervision, G.Z.M. All authors have read and agreed to the published version of the manuscript.

**Funding:** This research was carried out with the financial support of the Science Committee of the Ministry of Science and Higher Education of the Republic of Kazakhstan (BR21881822 Development of technological solutions for optimizing geological and technical operations when drilling wells and oil production at the late stage of field exploitation, 2023–2025).

**Institutional Review Board Statement:** Not applicable.

**Informed Consent Statement:** Not applicable.

**Data Availability Statement:** Data is contained within the article.

**Conflicts of Interest:** The authors declare no conflicts of interest.

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
