# Peer review of "Study of the Effectiveness of Corrosion Resistance Growth by Application of Layered AlN–TiO2 Coatings"

_coatings, doi:10.3390/coatings14040373_

Round 1

Reviewer 1 Report

Comments and Suggestions for Authors

The manuscript shows consistent and logical research of multilayered protective coatings. The research design is generally appropriate and obtained results are clearly presented, however, there are some specific comments and questions which are listed below.

1.     Introduction section should be revised. Now it is too general, more detailed overview is needed to show various materials for protective coatings including multilayered. Studies of AlN and TiO2 coatings also should be discussed since these materials are common and well-studied as protective ones. Moreover, combination of these materials was also proposed in various studies.

2.     Based on previous comment, the novelty of the study should be clarified.

3.     How was thickness of coatings layers measured/controlled? Cross-sectional micrographs are desirable.

4.     Are you sure that indenter didn’t reach the 316L during hardness tests? How can you prove it?

5.     Equivalence of the simulation of external atmospheric influences to 5-7 years of natural aging processes should be justified.

6.     Despite the given explanation and suggestions about the origin of the observed effects, more detailed analysis and discussion about mechanisms of properties (contact angle, hardness, critical load, wear resistance) reduction during corrosion and aging tests are needed. Given suggestions about chemical changes should be justified.

Author Response

Review Report Form 1

Introduction section should be revised. Now it is too general, more detailed overview is needed to show various materials for protective coatings including multilayered. Studies of AlN and TiO2 coatings also should be discussed since these materials are common and well-studied as protective ones. Moreover, combination of these materials was also proposed in various studies.

The authors thank the reviewer for this comment; the introduction has been expanded and a description of the use of various coatings as protective materials is presented. The introduction has been revised to take into account comments from all reviewers.

The choice of coatings based on AlN – TiO2 obtained by magnetron sputtering as objects for research is based on the combination of their properties, which allow their use as one of the heat-resistant protective coatings capable of operating at high temperatures. For example, the combination of strength and thermal insulation properties of AlN coatings, which makes it possible to increase the resistance of materials to corrosion processes, is presented in [25]. In [26], the use of multilayer AlN/Si made it possible to increase the corrosion resistance of 304 steel. The use of nanostructured TiO2 modified in various ways as anti-corrosion coatings is considered in [27,28]. Interest in layered coatings based on various compounds, including oxides and nitrides, in the case of their use as protective materials in recent years has been quite large. In particular, the use of layered coatings makes it possible to create barrier layers on the surface of steel structures that can reduce the rate of degradation. So, for example, the use of layered coatings based on CrNi, TiAlN/CrNi, CrNi–Al2O3–TiO2 and TiN/TiO2 as anti-corrosion coatings was proposed in [29,30]. Using these coatings, the authors managed to increase the wear resistance and corrosion resistance of materials. Much attention is also paid to the development of compounds such as TiO2/graphene or MXene/EP as protective coatings, which have great prospects when used as anticorrosion or antibacterial coatings [31,32]. In addition to the use of multilayer films or coatings as protective anti-corrosion materials, recently much attention has been paid to the use of these structures as cathode materials for batteries or fuels, due to their high resistance to external influences and degradation processes [33,34].

2.     Based on previous comment, the novelty of the study should be clarified

The authors thank the reviewer for this comment; novelty and relevance have been added.

Based on the above, the novelty of the presented research lies in determining the variation in the number of layers of AlN – TiO2 coatings for resistance to corrosion and aging, while maintaining the thickness of the coating. The use of layered coatings can be used not only to increase corrosion resistance, but also to strengthen the surface layers, as well as increase resistance to external influences.

3.     How was thickness of coatings layers measured/controlled? Cross-sectional micrographs are desirable.

The authors thank the reviewer for this comment; the data are given in the text of the article.

Figure 1e shows an example of a side cleavage of the resulting sprayed coatings, combined with mapping results that reflect the elemental composition of the resulting layers. Moreover, according to the obtained mapping data, the layers are compounds of aluminum with nitrogen and titanium with oxygen, which indicates the purity of the sprayed layers, as well as the absence of mixing effects between the layers.

4.     Are you sure that indenter didn’t reach the 316L during hardness tests? How can you prove it?

Hardness measurements were carried out on the basis of a priori experimental data related to determining the geometry of the indenter prints and their depth during indentation with different loads. The choice of load on the indenter was selected in such a way that the measurement was carried out only at a depth corresponding to the thickness of the coating, without pushing through the substrate.

5.     Equivalence of the simulation of external atmospheric influences to 5-7 years of natural aging processes should be justified.

These conditions for conducting aging experiments were chosen in accordance with previously conducted similar experiments related to the demonstration of artificial aging processes, which made it possible to compare the results obtained with the effects of atmospheric conditions on samples during their operation.

6.     Despite the given explanation and suggestions about the origin of the observed effects, more detailed analysis and discussion about mechanisms of properties (contact angle, hardness, critical load, wear resistance) reduction during corrosion and aging tests are needed. Given suggestions about chemical changes should be justified.

Figure 8 shows a schematic representation of the corrosion processes of coating samples when interacting with an aggressive environment. In the case of single-layer coatings, for which the formation of cavities characteristic of ulcer formations associated with interaction with Cl- ions was observed, degradation is accompanied by both the formation of growths and partial destruction of the surface. Moreover, for steel 316 L these processes proceed much faster due to the formation of oxides in the form of iron oxide, which leads not only to the formation of build-up, but also to surface degradation in the form of microchips and cracks. In the case of layered coatings, the presence of interlayer boundaries leads to inhibition of the processes of propagation of ulcer formations in depth, which in turn leads to the fact that the cavities formed on the surface of layered coatings have very small sizes.

Figure 8. Schematic representation of the corrosion processes of coating surfaces when interacting with an aggressive environment: a) single-layer coatings at the beginning of testing; b) single-layer coatings after testing; c) multilayer coatings at the beginning of testing; d) multi-layer coatings after testing

Reviewer 2 Report

Comments and Suggestions for Authors

The work aimed to study the use of AlN – TiO2 coatings as protective materials against corrosion and natural aging.  The corrosion resistance of steel structures, as well as increase wear resistance under long-term mechanical influences (friction and aging) were conducted. The work has potential applications in engineering. 

1) How was the thickness of the coating determined and as well as the each layer?

2) whether the final thickness and topography was detecked?

3) as the contact angle was mostly effected by profile and surface chemistry, why there is still a difference between the sample a, c and d?

4) in the introduction, the ref (doi.org/10.1016/j.porgcoat.2023.107779) could be cited.

5) the connecting between the corrosion and different layers should be reinforced.

Author Response

Review Report Form 2

1) How was the thickness of the coating determined and as well as the each layer?

The authors thank the reviewer for this comment. The following information has been added to the text of the article.

The thickness was determined using the ellipsometry method. The total thickness of the deposited coatings was about 600±10 nm. Figure 1 shows a schematic representation of the studied samples of AlN – TiO2 coatings obtained by magnetron sputtering. In total, 4 types of coating were obtained in the experiment: 1) a single-layer AlN coating with a thickness of about 600 nm; 2) single-layer TiO2 coating with a thickness of about 600 nm; 3) coating consisting of 3 layers of AlN and 3 layers of TiO2, each 100 nm thick; 4) coating consisting of 6 layers of AlN – 6 layers of TiO2, each 50 nm thick. Figure 1 shows a schematic representation of the resulting coatings. The choice of 6 and 12 deposition layers is determined by the capabilities of the magnetron installation, as well as the possibility of obtaining layers with a minimum thickness of 50 nm. At the same time, the main goal in selecting spraying conditions was the possibility of obtaining coatings with a thickness of about 0.5 – 0.6 μm with a multiple number of layers during spraying. The reasons for choosing a method for applying coatings are due to the possibility of scaling this coating method to sufficiently large volumes of sample surfaces. Figure 1e shows an example of a side cleavage of the resulting sprayed coatings, combined with mapping results that reflect the elemental composition of the resulting layers. Moreover, according to the obtained mapping data, the layers are compounds of aluminum with nitrogen and titanium with oxygen, which indicates the purity of the sprayed layers, as well as the absence of mixing effects between the layers.

2) whether the final thickness and topography was detecked?

The authors thank the reviewer for this comment. The following information has been added to the text of the article.

Figure 3 shows SEM images of the surface of coating and steel samples before and after corrosion tests.

3) as the contact angle was mostly effected by profile and surface chemistry, why there is still a difference between the sample a, c and d?

The authors thank the reviewer for this comment.

4) in the introduction, the ref (doi.org/10.1016/j.porgcoat.2023.107779) could be cited.

The authors thank the reviewer for this comment; the introduction has been expanded and a description of the use of various coatings as protective materials is presented. The introduction has been revised to take into account comments from all reviewers.

5) the connecting between the corrosion and different layers should be reinforced.

Figure 8 shows a schematic representation of the corrosion processes of coating samples when interacting with an aggressive environment. In the case of single-layer coatings, for which the formation of cavities characteristic of ulcer formations associated with interaction with Cl- ions was observed, degradation is accompanied by both the formation of growths and partial destruction of the surface. Moreover, for steel 316 L these processes proceed much faster due to the formation of oxides in the form of iron oxide, which leads not only to the formation of build-up, but also to surface degradation in the form of microchips and cracks. In the case of layered coatings, the presence of interlayer boundaries leads to inhibition of the processes of propagation of ulcer formations in depth, which in turn leads to the fact that the cavities formed on the surface of layered coatings have very small sizes.

Figure 8. Schematic representation of the corrosion processes of coating surfaces when interacting with an aggressive environment: a) single-layer coatings at the beginning of testing; b) single-layer coatings after testing; c) multilayer coatings at the beginning of testing; d) multi-layer coatings after testing

Reviewer 3 Report

Comments and Suggestions for Authors

Manuscript Number: Coatings 2894653

The manuscript written by Moldabayeva et al. titled “Study of the effectiveness of corrosion resistance growth by application of layered AlN – TiO2 coatings” provided a study where a multilayer coating system deposited via sputtering process was compared to the individual coating systems to protect metal substrate. I want to suggest a few edits and comments which will further improve the article and make it suitable for Coatings.

Comments

1.      The authors have assessed the corrosion protection with increase in the hydrophilicity of the coating surface after the exposure to the model solution. I provide a few pictures of the coatings before and after exposure to the model NaCl solution.

2.      Please provide surface roughness data before and after exposure as it can be correlated to the decrease in water contact angle.

3.      What the authors meant by decrease in strength characteristics? Please give some information about what properties the authors are referring to.

4.      Did the authors studies any increase in defects like blistering, delamination between the substrate and the coating system. If yes, please provide the results.

5.      Was there any EIS study carried out to assess the electrochemical resistance provided by different coating systems?

6.      Use of layer-by-layer deposition of the multilayer coating was carried out by Yoon et al. It would be advised to include the bellow studies-

https://doi.org/10.1016/j.ssi.2012.09.011; https://doi.org/10.1016/j.tsf.2012.09.027

Author Response

Review Report Form 3

1. The authors have assessed the corrosion protection with increase in the hydrophilicity of the coating surface after the exposure to the model solution. I provide a few pictures of the coatings before and after exposure to the model NaCl solution.

The authors thank the reviewer for this comment; images of the sample surfaces before and after corrosion tests have been added to the text of the article.

Figure 3 shows SEM images of the surface of coating and steel samples before and after corrosion tests. As can be seen from the presented data, the presence of 316 L steel in model solutions leads to the formation of growths on its surface in the form of salt deposits, as well as oxides. At the same time, for single-layer AlN and TiO2 coatings after corrosion tests, the formation of cavities is observed, indicating the processes of pitting or pitting corrosion, leading to partial destruction of the coating. In the case of layered coatings, such phenomena are not observed, and the main changes are associated only with the formation of small salt deposits on the surface.

2.      Please provide surface roughness data before and after exposure as it can be correlated to the decrease in water contact angle.

The authors thank the reviewer for this comment.

According to the presented data on the morphological features of the coatings under study in Figure 3, in the case of single-layer AlN coatings, the roughness value before testing is about 30 – 40 nm. while after testing this value increases to 70 -110 nm, and crater-like cavities are observed on the surface, with a depth of about 300 - 400 nm and a diameter of about 2 - 3 μm, which indicates ulcerative destruction of the coating. In the case of a TiO2 coating in the initial state, the roughness is about 20 – 30 nm, and after testing the roughness value is more than 70 nm, while on the surface, as in the case of AlN coatings, the formation of craters and build-ups on the surface of the coating is observed. For 3AlN – 3TiO2 coatings in the initial state, the roughness is about 40 – 50 nm, which is comparable to the observed values for AlN coatings. At the same time, for 3AlN – 3TiO2 coatings after corrosion tests, the roughness practically did not change and was about 45 – 55 nm. In the case of 6AlN – 6TiO2 coatings, a decrease in the thickness of the layers leads to the formation of smaller grains, and the roughness value is about 20 – 30 nm, while this value does not undergo significant changes after corrosion tests.

3.      What the authors meant by decrease in strength characteristics? Please give some information about what properties the authors are referring to.

The authors thank the reviewer for this comment.

In this case, a decrease in strength properties means a change in hardness and critical load at which the coating is torn off from the steel surface during testing.

4.      Did the authors studies any increase in defects like blistering, delamination between the substrate and the coating system. If yes, please provide the results.

The authors thank the reviewer for this comment.

The article presents the results of a study of the morphological features of the surface of coating samples after corrosion tests, according to which the formation of growths or partial destruction is observed in the case of single-layer coatings. At the same time, no effects of delamination or partial peeling from the surface were observed.

5.      Was there any EIS study carried out to assess the electrochemical resistance provided by different coating systems?

The authors thank the reviewer for this comment, however, similar measurements were not carried out for these samples, since the main goal of the work was to determine the strength characteristics of coatings and their changes as a result of corrosion tests. However, in the future we will definitely conduct similar studies for similar coating samples.

6. Use of layer-by-layer deposition of the multilayer coating was carried out by Yoon et al. It would be advised to include the bellow studies-

https://doi.org/10.1016/j.ssi.2012.09.011; https://doi.org/10.1016/j.tsf.2012.09.027

The authors thank the reviewer for this comment; the introduction has been expanded and a description of the use of various coatings as protective materials is presented. The introduction has been revised to take into account comments from all reviewers.

Reviewer 4 Report

Comments and Suggestions for Authors

The manuscipt is interesting but there are certain aspects to be considered

1. Literature on the magnetron sputtering works related to AlN-TiO2 coatings and the gaps that are being addressed is missing in the mansucript

2. Scanning electron micrographs showing different layers of AlN-TiO2 followed by thickness measurements is missing

3. AlN-TiO2 combination will work to enhance corrosion resistance, but why 6 layers? and why only magnetron sputtering?

4. Schematic showing the mechanism and control of corrosion is recommended to represent the summary of the discussion

5. 'From which it follows that the change in strength parameters has a clear dependence not only on the time spent in the model solution, an increase in which leads to destabilization of the near-surface layer due to migration processes of the introduction of Cl- ions into the coatings, with a subsequent increase in their concentration, which can lead to destabilization of strength parameters'- The sentence is too long to follow and not sure what it signifies. The authors need to rephrase, revise and emphasise on what they want to say.

6. Thickness data seems to be missing

7. Conclusion doesnt cover all the details mentioned in the manuscript and needs to be revised

Comments on the Quality of English Language

From which it follows that the change in strength parameters has a clear dependence not only on the time spent in the model solution, an increase in which leads to destabilization of the near-surface layer due to migration processes of the introduction of Cl- ions into the coatings, with a subsequent increase in their concentration, which can lead to destabilization of strength parameters- There are few sentences like this which are too long and confusing the readers. The authors need to rephrase and revise the text for the non-specialist audience to understand

Author Response

Review Report Form 4

1. Literature on the magnetron sputtering works related to AlN-TiO2 coatings and the gaps that are being addressed is missing in the mansucript

The authors thank the reviewer for this comment; the introduction has been expanded and a description of the use of various coatings as protective materials is presented.

The choice of coatings based on AlN – TiO2 obtained by magnetron sputtering as objects for research is based on the combination of their properties, which allow their use as one of the heat-resistant protective coatings capable of operating at high temperatures. For example, the combination of strength and thermal insulation properties of AlN coatings, which makes it possible to increase the resistance of materials to corrosion processes, is presented in [25]. In [26], the use of multilayer AlN/Si made it possible to increase the corrosion resistance of 304 steel. The use of nanostructured TiO2 modified in various ways as anti-corrosion coatings is considered in [27,28]. Interest in layered coatings based on various compounds, including oxides and nitrides, in the case of their use as protective materials in recent years has been quite large. In particular, the use of layered coatings makes it possible to create barrier layers on the surface of steel structures that can reduce the rate of degradation. So, for example, the use of layered coatings based on CrNi, TiAlN/CrNi, CrNi–Al2O3–TiO2 and TiN/TiO2 as anti-corrosion coatings was proposed in [29,30]. Using these coatings, the authors managed to increase the wear resistance and corrosion resistance of materials. Much attention is also paid to the development of compounds such as TiO2/graphene or MXene/EP as protective coatings, which have great prospects when used as anticorrosion or antibacterial coatings [31,32]. In addition to the use of multilayer films or coatings as protective anti-corrosion materials, recently much attention has been paid to the use of these structures as cathode materials for batteries or fuels, due to their high resistance to external influences and degradation processes [33,34]. 

2. Scanning electron micrographs showing different layers of AlN-TiO2 followed by thickness measurements is missing

Figure 1e shows an example of a side cleavage of the resulting sprayed coatings, combined with mapping results that reflect the elemental composition of the resulting layers. Moreover, according to the obtained mapping data, the layers are compounds of aluminum with nitrogen and titanium with oxygen, which indicates the purity of the sprayed layers, as well as the absence of mixing effects between the layers.

3. AlN-TiO2 combination will work to enhance corrosion resistance, but why 6 layers? and why only magnetron sputtering?

The choice of 6 deposition layers is determined by the capabilities of the magnetron installation, as well as the possibility of obtaining layers with a minimum thickness of 50 nm. At the same time, the main goal in selecting spraying conditions was the possibility of obtaining coatings with a thickness of about 0.5 – 0.6 μm with a multiple number of layers during spraying. The reasons for choosing a method for applying coatings are due to the possibility of scaling this coating method to sufficiently large volumes of sample surfaces.

4. Schematic showing the mechanism and control of corrosion is recommended to represent the summary of the discussion

Figure 8 shows a schematic representation of the corrosion processes of coating samples when interacting with an aggressive environment. In the case of single-layer coatings, for which the formation of cavities characteristic of ulcer formations associated with interaction with Cl- ions was observed, degradation is accompanied by both the formation of growths and partial destruction of the surface. Moreover, for steel 316 L these processes proceed much faster due to the formation of oxides in the form of iron oxide, which leads not only to the formation of build-up, but also to surface degradation in the form of microchips and cracks. In the case of layered coatings, the presence of interlayer boundaries leads to inhibition of the processes of propagation of ulcer formations in depth, which in turn leads to the fact that the cavities formed on the surface of layered coatings have very small sizes.

Figure 8. Schematic representation of the corrosion processes of coating surfaces when interacting with an aggressive environment: a) single-layer coatings at the beginning of testing; b) single-layer coatings after testing; c) multilayer coatings at the beginning of testing; d) multi-layer coatings after testing

5. 'From which it follows that the change in strength parameters has a clear dependence not only on the time spent in the model solution, an increase in which leads to destabilization of the near-surface layer due to migration processes of the introduction of Cl- ions into the coatings, with a subsequent increase in their concentration, which can lead to destabilization of strength parameters'- The sentence is too long to follow and not sure what it signifies. The authors need to rephrase, revise and emphasise on what they want to say.

The authors thank the reviewer for this comment; the sentence has been paraphrased.

According to the data obtained, it was established that the change in strength parameters has a clear dependence on the time spent in the model solution. At the same time, an increase in the experiment time leads to destabilization of the near-surface layer due to migration processes of the introduction of Cl- ions into the coatings with a subsequent increase in their concentration, leading to a deterioration in strength properties.

6. Thickness data seems to be missing

The thickness was determined using the ellipsometry method. The total thickness of the deposited coatings was about 600±10 nm. Figure 1 shows a schematic representation of the studied samples of AlN – TiO2 coatings obtained by magnetron sputtering. In total, 4 types of coating were obtained in the experiment: 1) a single-layer AlN coating with a thickness of about 600 nm; 2) single-layer TiO2 coating with a thickness of about 600 nm; 3) coating consisting of 3 layers of AlN and 3 layers of TiO2, each 100 nm thick; 4) coating consisting of 6 layers of AlN – 6 layers of TiO2, each 50 nm thick.

7. Conclusion doesnt cover all the details mentioned in the manuscript and needs to be revised

The authors thank the reviewer for this comment. The conclusion has been expanded and supplemented.

As a result of modeling the aging processes, it was found that the use of layered coatings can significantly reduce the rate of natural aging of the surface, as well as maintain strength parameters practically unchanged.

The obtained experimental results will be used in the future in the development of technological solutions related to increasing the service life of steel structures operating in difficult conditions, including long-term interaction with aggressive environments.

Round 2

Reviewer 1 Report

Comments and Suggestions for Authors

Revision of the manuscript was carried out thoroughly, I'm satisfied with given answers.

There is only one comment left:

talking about simulate external atmospheric influences, corresponding reference should be given on "previously conducted similar experiments" where one can find detailed description.

Reviewer 2 Report

Comments and Suggestions for Authors

The author has already addressed all comments and could be acceptable.